# The Influence of Different Irrigation Scenarios on the Yield and Sustainability of Wheat Fodder under Hydroponic Conditions

Andrius Grigas *, Dainius Savickas , Dainius Steponavičius, Žygimantas Niekis and Jonas Balčiūnas

Department of Agricultural Engineering and Safety, Vytautas Magnus University Agriculture Academy, Studentų St. 15A, LT-53362 Akademija, Kaunas District, Lithuania
* Correspondence: andrius.grigas@vdu.lt

**Abstract:** Agriculture uses more water than any other resource to produce animal feed and wastes much of it through inefficiency. One possible alternative to solve this problem is hydroponically grown animal fodder, which under hydroponic conditions can achieve optimal results and, most importantly, use expensive resources, such as water, more efficiently. In the conducted research, different irrigation scenarios (IR1–IR6) were investigated when the water flow rate, irrigation frequency, and duration (IR1—1 l min$^{-1}$, 4 times day$^{-1}$, 120 s; IR2—2 l min$^{-1}$, 4 times day$^{-1}$, 120 s; IR3—3 l min$^{-1}$, 4 times day$^{-1}$, 120 s; IR4—1 l min$^{-1}$, 8 times day$^{-1}$, 60 s; IR5—2 l min$^{-1}$, 8 times day$^{-1}$, 60 s; and IR6—3 l min$^{-1}$, 8 times day$^{-1}$, 60 s) were changed during the hydroponic wheat fodder cultivation using a 7-day growth cycle. The results showed that the highest yield from the used seed was obtained in scenarios IR5 (5.95 ± 0.14 kg kg$^{-1}$) and IR6 (5.91 ± 0.19 kg kg$^{-1}$). In terms of frequency and irrigation duration, in IR1, IR2, and IR3, the average yield reached 4.7 ± 1.85 kg kg$^{-1}$, and in scenarios IR4, IR5, and IR6, the average yield was 15.4% higher—5.55 ± 1.63 kg kg$^{-1}$. The results obtained showed that by increasing the flow rate (from 1 l min$^{-1}$ to 3 l min$^{-1}$) and the frequency of irrigation (from 4 times day$^{-1}$ to 8 times day$^{-1}$), the yield increased by 32.5%, but the mass of the grown fodder per liter of water used simultaneously decreased by 50.6%. The life cycle assessment showed that although irrigation scenario IR4 had the most efficient use of water, the $CO_2$ footprint per functional unit (FU) was the highest due to the lowest yield compared to the other five irrigation scenarios. The lowest environmental impacts per FU were obtained in scenarios IR5 and IR6 (100.5 ± 3.3 and 100.6 ± 2.4 kg $CO_{2eq}$ t$^{-1}$, respectively).

**Keywords:** environmental impact; hydroponic; irrigation; LCA; nutrient film technique; wheat fodder



## 1. Introduction

Agriculture has a significant impact on the environment [1,2]. Traditional soil-based agriculture is increasingly facing challenges such as the reduction in and degradation of arable land areas and the lack of freshwater worldwide [3]. Agriculture, which produces animal feed, consumes the most water of all other resources, and most of it is wasted due to inefficient use [4]. Approximately 70 percent of global freshwater is used in agriculture [5], and a large share of it goes into feed and livestock production [6]. Conventional methods of intensification of agriculture and the particularly high use of irrigation and fertilizers are some of the main components that have a negative impact on the environment [7]. To mitigate negative impacts, it is important to adopt more sustainable agricultural practices, such as promoting the conservation and management of natural resources and developing more sustainable food and feed production systems.

Hydroponic systems offer a great opportunity to produce animal feed under controlled conditions in a more sustainable way and increase the efficiency of the resources used. In terms of water usage, hydroponic systems can be relatively efficient [8]. This is because the water used for growing plants is constantly recycled and only a small amount is lost to evaporation. Additionally, because the growth cycles are relatively short, the system

requires little water compared to traditional fodder crops. It has been proven that producing 1 kg of fodder using traditional methods requires an average of 85 L of water, [9] while producing the same amount of fresh hydroponic fodder requires only about 3 L of water [9]. Hydroponic fodder production is a method of growing sprouted grains and legumes as feed for livestock [6]. Using this system, seeds are sprouted in water, usually in trays or troughs, and then fed to animals as a fresh and nutritious source of fodder [10]. Hydroponic fodder is nutritionally superior to conventional concentrates in terms of crude protein, organic matter, ether extract, and nitrogen-free extracts [11]. Using the NFT (Nutrient Film Technique)—the most widely used irrigation fluid delivery system—fodder can be grown in 7 days [12]. Using this system, the growth tray is placed at an angle to allow the irrigation fluid to flow down toward the bottom, and a fresh fluid is being pumped into the highest point of the tray. It should also be emphasized that when growing fodder using a 7-day growth cycle, additional fertilization is not required, which has a positive impact on the use of resources [12]. Scientists have studied various types of cereals, such as wheat [12–15], barley [10,16], and maize [17,18], in the cultivation of hydroponic fodder, and the benefits of such feed for animals have been proven. However, there is a lack of data on the influence of various cultivation parameters on fodder yield and environmental impact, especially regarding the role of water and irrigation scenarios for efficient fodder production.

One of the most important hydroponic growth parameters that influence the growth of fodder is irrigation. It is the process of delivering water to plants to meet their needs for several important processes [19]. It is known that providing too little or too much water can reduce crop productivity or, when extreme, can lead to plant death [20], especially during germination. Water triggers a series of biochemical reactions that initiate the growth of a new plant [21]. It is needed to moisten the metabolic activity of the protoplasm, supply dissolved $O_2$ to the seed embryo, soften the outer seed coat, and improve the permeability of the seed [22]. In the absence of water, enzymatic activity decreases, which negatively affects carbohydrate metabolism and water potential; in addition, the amount of soluble calcium and potassium decreases, and seed hormones change. [23,24]. The amount of water required for germination may vary depending on the type of seed and environmental conditions. In general, the seeds should be kept moist, but not soaked, as too much water can cause rotting and fungal growth. The timing and frequency of irrigation can also affect germination. Optimal frequent irrigation can help initiate the germination process, while delaying watering can reduce germination [25].

Water is an essential factor in seed germination [26] and must be carefully managed to ensure that the seeds have sufficient moisture to initiate growth and that they do not become waterlogged or exposed to harmful contaminants. The choice of the best irrigation scenario for hydroponic wheat fodder production depends on the needs and objectives of the producer. A comprehensive evaluation of the different irrigation scenarios is needed to form recommendations for hydroponic fodder producers when developing an irrigation strategy, to not only achieve better economic indicators of the farm but also to contribute to a more sustainable use of expensive resources. Therefore, the present study aims to investigate the influence of different irrigation scenarios on the yield and sustainability of wheat fodder under hydroponic conditions using a 7-day growth cycle. The following scientific hypotheses were tested: (1) a higher frequency of Irrigation with a lower flow rate and duration will have the most effect on the yield of hydroponic fodder, and (2) fodder grown under these conditions will have the lowest $CO_2$ footprint per functional unit.

## 2. Materials and Methods

Scientific research was conducted from 2022 to 2023 at the Agricultural Machines Technological Process Research Laboratory of Vytautas Magnus University Agriculture Academy, Lithuania, using a self-designed and manufactured hydroponic fodder cultivation and research bench (Figure 1).

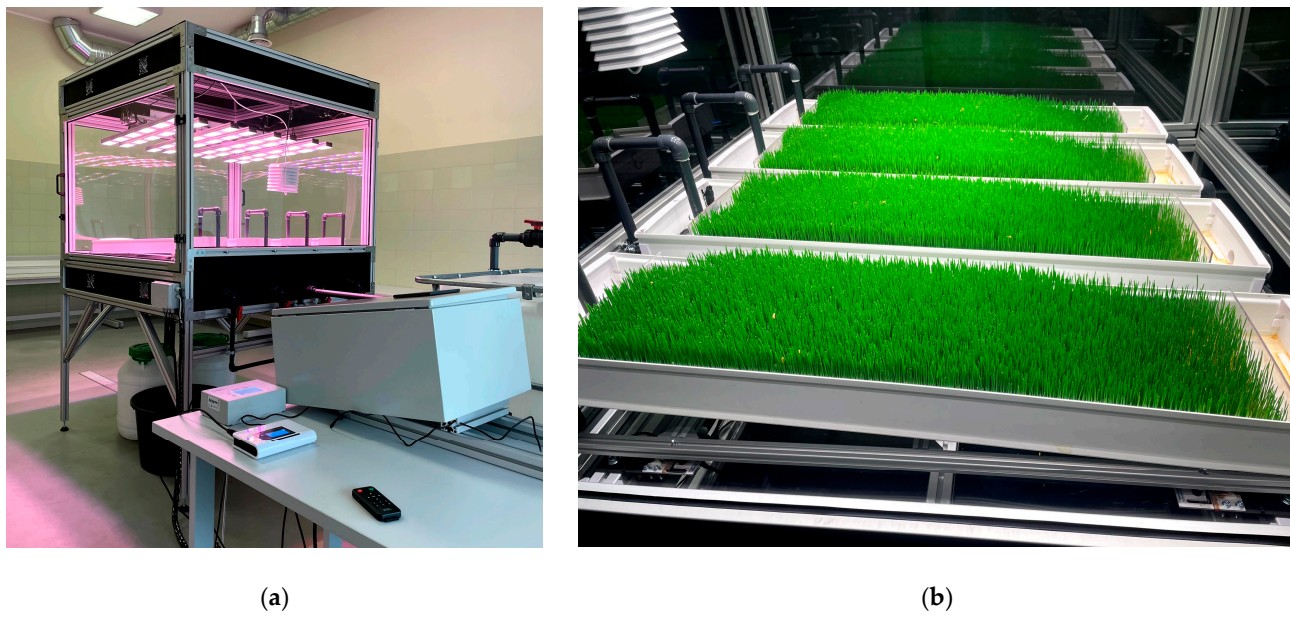

**(a)**                    **(b)**

**Figure 1.** (**a**) Hydroponic fodder cultivation and research bench; (**b**) hydroponic wheat fodder on the sixth day of cultivation.

Its frame (dimensions of which are 2150 × 1400 × 1400 mm) was made using extruded aluminum profiles (2). The inside of the bench is equipped with a fodder growth and weighing unit with four growth trays (7), each with a length of 1000 mm, a width of 225 mm, and a height of 70 mm (Figure 2). One growth tray has a cultivation area of 0.9 m². NFT was used as the irrigation fluid delivery system.

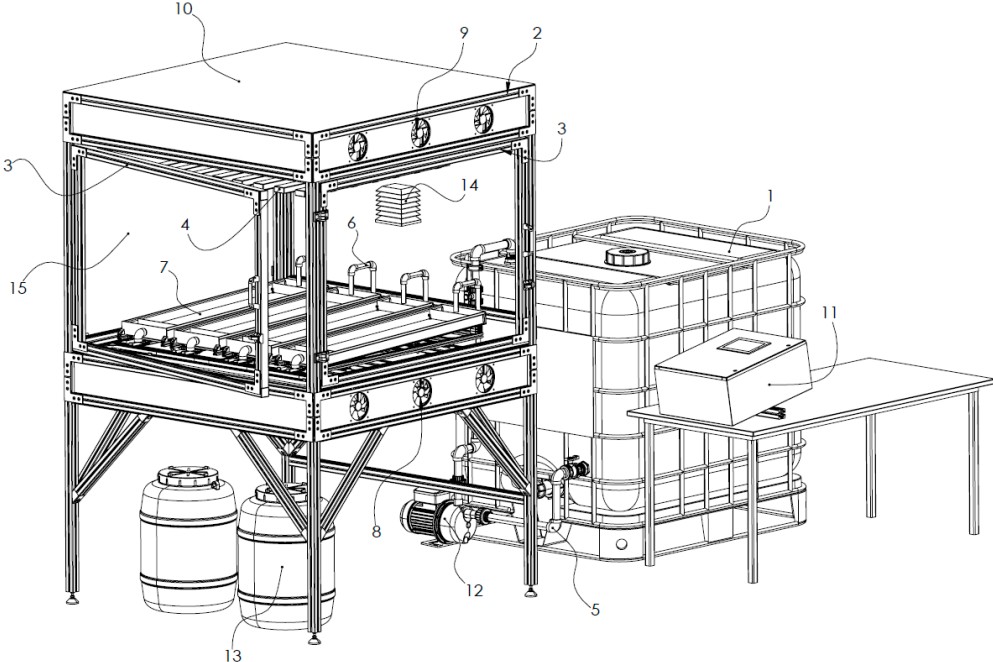

**Figure 2.** Hydroponic fodder cultivation and research bench scheme: 1—irrigation tank, 2—aluminum profile, 3—door, 4—aluminum frame with LED system installed, 5 and 6—irrigation piping and connections, 7—growth tray unit equipped with a system for monitoring changes in the mass of cultivated plants, 8—air intake system, 9—exhaust air ventilation system, 10—plastic cover plate, 11—process control unit, 12—irrigation pump, 13—seed soaking container, 14—measurement unit with microclimate sensors installed in it, 15—plexiglass panels.

### 2.1. Preparation for Fodder Cultivation

Before each growth cycle, the desired temperature and relative humidity threshold values (20 °C for temperature and 40% for humidity with a hysteresis of 1 °C and 3%, respectively) of the bench were set in the software. Since the slope angle of the growth tray is particularly significant [12], during all tests the growth trays were tilted at an angle of 6.5% using calibrated plastic spacers (Figure 3) (one spacer of 10 mm corresponds to a 1% slope angle). The slope angle was verified using Bosch DNM 60 L electronic spirit level with an accuracy of $\pm$ 0.05°. In addition, using the valves of the irrigation pipeline (located at the back of the bench), the required flow rate of the irrigation fluid was determined, and the irrigation frequency was determined in the irrigation controller located in the control unit (11). Then, with the help of the controller, the desired 12 h photoperiod, the visible spectrum of light (400–700 nm), and the photosynthetic flux density of 250 $\mu$mol m$^{-2}$ s$^{-1}$ for each growth tray were set.

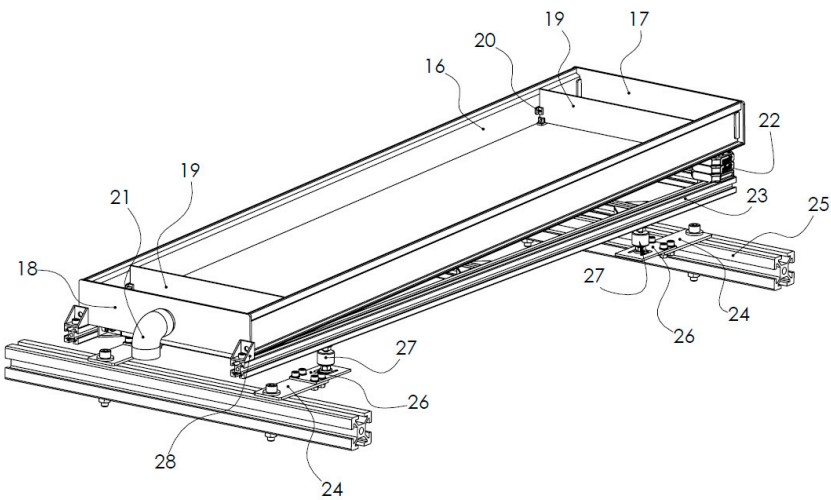

**Figure 3.** Growth tray unit: 16—growth tray, 17—growth tray's back cover, 18—growth tray's front cover, 19—a barrier preventing the washing out of grains, 20—grain barrier holders, 21—outlet, 22—plastic-calibrated 10 mm spacers, 23—aluminum profile, 24—bracket for mounting the load cell to the bench frame, 25—part of the bench structure, 26—load cells, 27—holders, 28—growth tray limiter to prevent sliding.

### 2.2. Seeds and Their Preparation

The selection of seeds used for hydroponic fodder production depends on availability and geographical and agroclimatic conditions. Wheat is the most widely cultivated crop in the Baltic region of Europe and should therefore be used for hydroponic cultivation. Winter wheat (*Triticum aestivum L.*) seeds of the Tobak variety were used for the research, with a weight of 1000 grains of 48.5 g. The seeds were produced on a farm in Kėdainiai District, Lithuania. The seeds were cleaned and germinated on moistened filter paper [27], resulting in 98.6 $\pm$ 1.4% germination. Growth trays were cleaned and disinfected using a 20% sodium hypochlorite (NaOCl) solution. The cleaned wheat seeds were disinfected in a 3% hydrogen peroxide (H$_2$O$_2$) solution for 15 min, and then the seeds were rinsed and left to soak for 12 h before seeding [28]. Since tap water was used for seed soaking and irrigation, it is worth emphasizing that 94.4% of drinking water from the centralized drinking water supply in Lithuania meets the requirements of drinking water standard HN 24:2003 and is safe for consumption [29]. The water temperature during soaking was 19.87 $\pm$ 0.15 °C. Then, a uniform 30 mm thick layer of soaked grain weighing 1.55 kg (1.00 $\pm$ 0.03 kg dry grain before soaking) was spread on each growth tray.

### 2.3. Hydroponic Wheat Fodder Growth Conditions

#### 2.3.1. Irrigation

In this study, 6 different irrigation scenarios were investigated when the flow rate and the frequency of irrigation were changed during the hydroponic wheat fodder cultivation using a 7-day growth cycle. The first three scenarios had the same irrigation frequency and duration—4 times day$^{-1}$ for 2 min each—but different water flow rates: IR1—1 l min$^{-1}$, IR2—2 l min$^{-1}$, and IR3—3 l min$^{-1}$. The remaining three also had the same frequency and duration, but they differed from the first three scenarios, 8 times day$^{-1}$ for 1 min each, but different flow rates: IR4—1 l min$^{-1}$, IR5—2 l min$^{-1}$, and IR6—3 l min$^{-1}$. Thus, the total amount of water used per day was the same between IR1 and IR4, between IR2 and IR5, and between IR3 and IR6 (Table 1).

**Table 1.** Different irrigation scenarios and their parameters.

| Parameters | IR1 | IR2 | IR3 | IR4 | IR5 | IR6 |
|---|---|---|---|---|---|---|
| Irrigation frequency, times day$^{-1}$ | 4 | 4 | 4 | 8 | 8 | 8 |
| Flow rate, l min$^{-1}$ | 1 | 2 | 3 | 1 | 2 | 3 |
| Irrigation duration, s | 120 | 120 | 120 | 60 | 60 | 60 |
| Water amount per day, l | 8 | 16 | 24 | 8 | 16 | 24 |

Water was supplied to the highest point of the growth tray and due to potential energy differences, the irrigation fluid passes through the seeds or roots (depending on the day of germination). To ensure uniform conditions, after the irrigation process has been completed, the water is discharged into the sewer and is no longer reused.

The irrigation fluid flow rate was controlled using QTDS-100H Handheld ultrasonic water flow meter (Q&T Instrument, China) with an accuracy of ± 1% and set to the required value using hand-operated valves. Since it has been proven that when wheat fodder is grown hydroponically, fertilizers have no significant effect on yield [12], tap water with an electrical conductivity of 0.16 ± 0.03 mS cm$^{-1}$ was used for this study. The measuring device HI98192 (UAB Hanna Instruments Baltics, Lithuania), with a measurement accuracy of 0.01 mS cm$^{-1}$, was used to determine the electrical conductivity.

#### 2.3.2. Microclimate

This study was conducted under controlled and monitored temperature and humidity conditions (average temperature 20.8 ± 1.2 °C and humidity 42.0 ± 4.1%) (Figure 4).

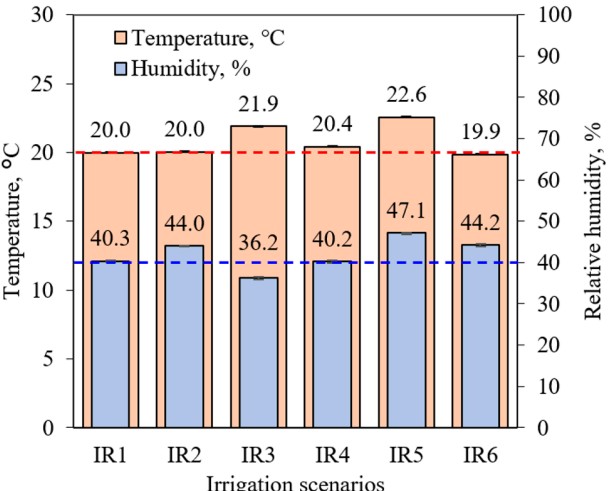

**Figure 4.** Temperatures and relative humidity over one growth cycle in each irrigation scenario. The red dashed line represents the set temperature value, and the blue one represents the humidity value.

The temperature in the growth chamber was controlled using the recuperation system in the room, and the humidity was controlled by the air intake and extraction system installed on the bench (8 and 9 in Figure 2).

### 2.3.3. Evaluation Criteria

Yield determination. The averages of the total yield of the germinated seeds were calculated using the following equation:

$$Y_t = \left( \sum_{i-1}^{n} (Wt_i - WS_i - W_i) \right)^{-n} \cdot k_i \tag{1}$$

where:

$Yt$ is the total yield of the germinated seeds, kg kg$^{-1}$;
$Wt$ is the total weight of wheat fodder, kg;
$WS$ is the weight of soaked wheat seeds that were spread in the growth tray, kg;
$W$ is the weight of the empty growth tray, kg;
$n$ is the number of repetitions, ($n$ = 4);
$k_i$ is the coefficient of imbibition, which was calculated as follows:

$$k_i = \frac{WS_d}{WS_{imb}} \tag{2}$$

where:

$WS_d$ is the weight of total dry wheat seeds that were soaked, kg;
$WS_{imb}$ is the weight of wheat seeds after soaking, kg.

Average weight gain per liter of water used. After the entire cultivation cycle, the average increase in weight of fodder per 1 L of water used was calculated, using the following equation:

$$W_g = \frac{Y_t}{V_w} \tag{3}$$

where:

$W_g$ is the average weight gain per liter of water used, g kg$^{-1}$;
$Y_t$ is the total yield of the germinated seeds, kg kg$^{-1}$;
$V_w$ is the volume of water used for the growth cycle, l.

### 2.4. Life Cycle Assessment

To assess the environmental impact of different irrigation scenarios, the life cycle assessment (LCA) method was used [30,31]. It is a widely used tool to assess the potential environmental impact and resource use of a product/service system throughout its life cycle, i.e., from raw material extraction to production and use stages to waste management and transportation [32], and is standardized by the International Commission of Standardization ISO 14040 [33] and ISO 14044 [34]. In this research, to perform LCA, SimaPro 9 software was used. Environmental impact assessment in the production of hydroponic wheat fodder was studied based on their midpoint impacts (CML-IA baseline V3.06/EU25), using 11 impact categories for evaluation (Table 2). Transportation, equipment, and raw materials (Table 3) were obtained from the Ecoinvent V3 database [35].

#### Scope and System Boundaries

The main goal of this environmental assessment is to determine the influence of different irrigation scenarios on the sustainability of wheat fodder under hydroponic conditions. For this goal, a theoretical hydroponic fodder farm located in the central part of Lithuania was analyzed and adapted to the climate and location and can therefore be used for comparison with local agricultural data relevant to the region and its climate. An insulated production building with a footprint of 100 m$^2$, equipped with 10 racks with a capacity of 800 growth trays (80 trays of 1000 × 225 × 75 mm in each rack), was chosen as

the research object. Each rack consisted of 5 vertical levels, each of which held 16 growth racks. Each level of the rack is equipped with LED lighting capable of ensuring a 12 h photoperiod with a luminous flux of 250 µmol m$^{-2}$ s$^{-1}$ of full spectrum (400–700 nm.). The boundary (Figure 5) of this LCA system includes everything needed to grow hydroponic wheat fodder using a 7-day growth cycle. The functional unit (FU), which is the reference unit for expressing environmental interventions, was expressed as 1 ton of wheat fodder grown using a hydroponic system.

**Table 2.** Selected impact categories, their abbreviations, and the measurement units.

| Impact Category | Abbreviation | Unit |
|---|---|---|
| Marine aquatic ecotoxicity | MAE | kg 1,4-DB$_{eq}$ |
| Abiotic depletion (fossil fuels) | ADf | MJ |
| Global warming | GWP | kg CO$_{2eq}$ |
| Freshwater aquatic ecotoxicity | FWAe | kg 1,4-DB$_{eq}$ |
| Human toxicity | HT | kg 1,4-DB$_{eq}$ |
| Terrestrial ecotoxicity | TE | kg 1,4-DB$_{eq}$ |
| Eutrophication | ET | kg PO$_{4eq}^{3}$ |
| Acidification | ACD | kg SO$_{2eq}$ |
| Photochemical oxidation | PO | kg C$_2$H$_{4eq}$ |
| Abiotic depletion | and | kg Sb$_{eq}$ |
| Ozone layer depletion | ODP | kg CFC-11$_{eq}$ |

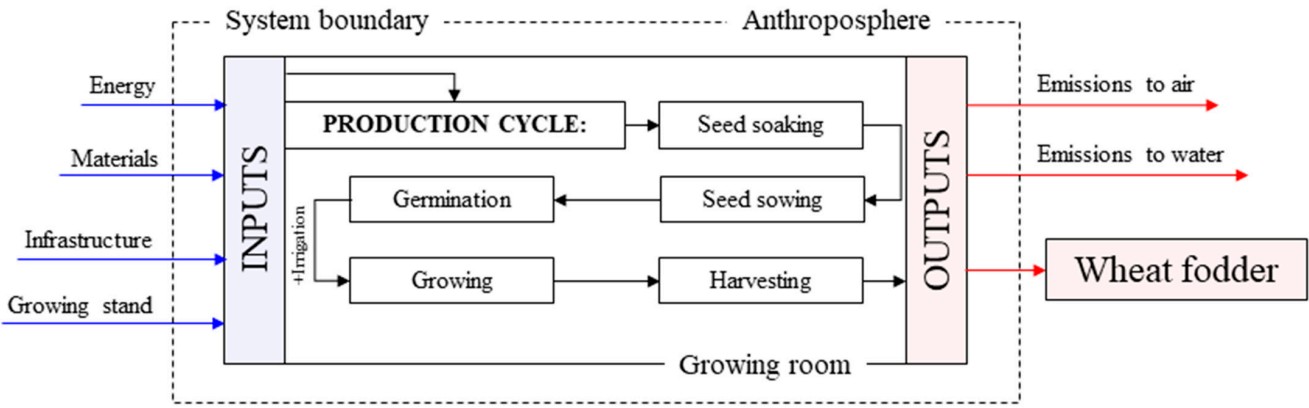

**Figure 5.** Flowchart and system boundary of the life cycle assessment of a hydroponic wheat fodder production facility.

As in the empirical yield study, in this life cycle assessment, different irrigation scenarios were studied (Table 1), so the yield results obtained in the test bench (Figure 1) were used to assess the environmental sustainability of the wheat fodder.

*2.5. Statistical Analysis*

All experiments in this study were repeated four times. Differences among means were compared by the one-way analysis of variance (ANOVA) module of the statistical software Statistica 10.0 and by performing Tukey HSD tests. A probability level of 0.05 was used as the criterion for tests of significance throughout the data analysis [36].

**Table 3.** Life cycle inventory of a hydroponic system operating for 30 days.

| Items | IR1 | | IR2 | | IR3 | | IR4 | | IR5 | | IR6 | |
|---|---|---|---|---|---|---|---|---|---|---|---|---|
| | Total | Per FU | Total | Per FU | Total | Per FU | Total | Per FU | Total | Per FU | Total | Per FU |
| **Tap water** | | | | | | | | | | | | |
| Seed soaking, $m^3$ | 1.2 | 0.082 | 1.2 | 0.072 | 1.2 | 0.062 | 1.2 | 0.071 | 1.2 | 0.057 | 1.2 | 0.057 |
| Irrigation, $m^3$ | 48.0 | 3.3 | 96.0 | 5.8 | 144.0 | 7.4 | 48.0 | 2.8 | 96.0 | 4.6 | 144.0 | 6.9 |
| **Electricity** | | | | | | | | | | | | |
| Heating, kWh | 1416.0 | 97.2 | 1416.0 | 85.2 | 1416.0 | 73.0 | 1416.0 | 83.6 | 1416.0 | 67.8 | 1416.0 | 67.4 |
| Lighting, kWh | 2160.0 | 148.2 | 2160.0 | 129.9 | 2160.0 | 111.4 | 2160.0 | 127.5 | 2160.0 | 103.5 | 2160.0 | 102.8 |
| Air movement, kWh | 54.0 | 3.7 | 54.0 | 3.2 | 54.0 | 2.8 | 54.0 | 3.2 | 54.0 | 2.6 | 54.0 | 2.6 |
| Dehumidifier, kWh | 144.0 | 9.9 | 144.0 | 8.7 | 144.0 | 7.4 | 144.0 | 8.5 | 144.0 | 6.9 | 144.0 | 6.9 |
| Irrigation, kWh | 8.4 | 0.6 | 8.4 | 0.5 | 8.4 | 0.4 | 8.4 | 0.5 | 8.4 | 0.4 | 8.4 | 0.4 |
| **Raw materials** | | | | | | | | | | | | |
| Wheat seeds, kg | 3531.43 | 242.4 | 3531.43 | 212.4 | 3531.43 | 182.2 | 3531.43 | 208.5 | 3531.43 | 169.2 | 3531.43 | 168.1 |
| Sodium hypochlorite, kg | 49.98 | 3.4 | 49.98 | 3.0 | 49.98 | 2.6 | 49.98 | 3.0 | 49.98 | 2.4 | 49.98 | 2.4 |
| **Transportation** | | | | | | | | | | | | |
| Seeds, km | 1.2 | 0.1 | 1.2 | 0.1 | 1.2 | 0.1 | 1.2 | 0.1 | 1.2 | 0.1 | 1.2 | 0.1 |
| **Construction** [a] | | | | | | | | | | | | |
| Growing stands | | | | | | | | | | | | |
| Aluminium, kg | 1.64 | 0.11 | 1.64 | 0.10 | 1.64 | 0.08 | 1.64 | 0.10 | 1.64 | 0.08 | 1.64 | 0.08 |
| Polyvinyl chloride PVC, kg | 19.56 | 1.34 | 19.56 | 1.18 | 19.56 | 1.01 | 19.56 | 1.15 | 19.56 | 0.94 | 19.56 | 0.93 |
| Lighting | | | | | | | | | | | | |
| Alluminium cast alloy, kg | $1.2 \times 10^{-1}$ | $8.4 \times 10^{-3}$ | $1.2 \times 10^{-1}$ | $7.4 \times 10^{-3}$ | $1.2 \times 10^{-1}$ | $6.3 \times 10^{-3}$ | $1.2 \times 10^{-1}$ | $7.2 \times 10^{-3}$ | $1.2 \times 10^{-1}$ | $5.9 \times 10^{-3}$ | $1.2 \times 10^{-1}$ | $5.8 \times 10^{-3}$ |
| Steel, kg | $2.6 \times 10^{-3}$ | $1.8 \times 10^{-4}$ | $2.6 \times 10^{-3}$ | $1.5 \times 10^{-4}$ | $2.6 \times 10^{-3}$ | $1.3 \times 10^{-4}$ | $2.6 \times 10^{-3}$ | $1.5 \times 10^{-4}$ | $2.6 \times 10^{-3}$ | $1.2 \times 10^{-4}$ | $2.6 \times 10^{-3}$ | $1.2 \times 10^{-4}$ |
| Copper, kg | $2.4 \times 10^{-4}$ | $1.6 \times 10^{-5}$ | $2.4 \times 10^{-4}$ | $1.4 \times 10^{-5}$ | $2.4 \times 10^{-4}$ | $1.2 \times 10^{-5}$ | $2.4 \times 10^{-4}$ | $1.4 \times 10^{-5}$ | $2.4 \times 10^{-4}$ | $1.1 \times 10^{-5}$ | $2.4 \times 10^{-4}$ | $1.1 \times 10^{-5}$ |
| Polyvinyl chloride PVC, kg | $9.5 \times 10^{-3}$ | $6.5 \times 10^{-4}$ | $9.5 \times 10^{-3}$ | $5.7 \times 10^{-4}$ | $9.5 \times 10^{-3}$ | $4.9 \times 10^{-4}$ | $9.5 \times 10^{-3}$ | $5.6 \times 10^{-4}$ | $9.5 \times 10^{-3}$ | $4.5 \times 10^{-4}$ | $9.5 \times 10^{-3}$ | $4.5 \times 10^{-4}$ |
| Glass, kg | $9.4 \times 10^{-3}$ | $6.4 \times 10^{-4}$ | $9.4 \times 10^{-3}$ | $5.6 \times 10^{-4}$ | $9.4 \times 10^{-3}$ | $4.8 \times 10^{-4}$ | $9.4 \times 10^{-3}$ | $5.5 \times 10^{-4}$ | $9.4 \times 10^{-3}$ | $4.5 \times 10^{-4}$ | $9.4 \times 10^{-3}$ | $4.5 \times 10^{-4}$ |
| Irrigation | | | | | | | | | | | | |
| Polyvinyl chloride PVC, kg | $2.7 \times 10^{-1}$ | $1.8 \times 10^{-2}$ | $2.7 \times 10^{-1}$ | $1.6 \times 10^{-2}$ | $2.7 \times 10^{-1}$ | $1.4 \times 10^{-2}$ | $2.7 \times 10^{-1}$ | $1.6 \times 10^{-2}$ | $2.7 \times 10^{-1}$ | $1.3 \times 10^{-2}$ | $2.7 \times 10^{-1}$ | $1.3 \times 10^{-2}$ |
| Cast iron, kg | $1.2 \times 10^{-1}$ | $8.4 \times 10^{-3}$ | $1.2 \times 10^{-1}$ | $7.4 \times 10^{-3}$ | $1.2 \times 10^{-1}$ | $6.3 \times 10^{-3}$ | $1.2 \times 10^{-1}$ | $7.2 \times 10^{-3}$ | $1.2 \times 10^{-1}$ | $5.9 \times 10^{-3}$ | $1.2 \times 10^{-1}$ | $5.8 \times 10^{-3}$ |
| El. Motor 2,4 kW, piece | $1.6 \times 10^{-2}$ | $1.1 \times 10^{-3}$ | $1.6 \times 10^{-2}$ | $9.8 \times 10^{-4}$ | $1.6 \times 10^{-2}$ | $8.4 \times 10^{-4}$ | $1.6 \times 10^{-2}$ | $9.7 \times 10^{-4}$ | $1.6 \times 10^{-2}$ | $7.8 \times 10^{-4}$ | $1.6 \times 10^{-2}$ | $7.8 \times 10^{-4}$ |
| **Outputs** | | | | | | | | | | | | |
| Wheat fodder, t | **14.57** | | **16.63** | | **19.39** | | **16.94** | | **20.88** | | **21.01** | |

[a]—adjusted for lifetime.

## 3. Results and Discussion

### 3.1. Wheat Fodder Yield Results

In our comparative studies of growing wheat fodder under hydroponic conditions, the influence of six different irrigation scenarios (Table 1) was analyzed. The conducted empirical studies helped to confirm the hypothesis that more frequent irrigation with a lower flow rate and duration will have the greatest impact on fodder yield grown using hydroponic technology.

A growing and actively functioning plant consists of 90–95% water. The remaining dry matter is varyingly composed and consists of organic compounds accounting for about 80 to 90%: carbohydrates and proteins with amino acids next to the minerals [37]. Due to the high water content of the plants, they are particularly vulnerable to water stress and drought conditions. When comparing the amount of water contained in the fresh matter of a given plant to the total amount of water taken up by the plant in its roots, only a small fraction (about 1%) of the total water enters the fresh matter [38]. The rest of the water is evaporated to lower the temperature of the plant. The potential rate of water removal by plants from the root zone is determined by several parameters: temperature, light, relative humidity, wind speed, leaf area, and the degree to which the plant's stomata are open [39].

In this study, it was found that the highest yield, after 7 days of cultivation, was in wheat fodder that was irrigated using IR5 and IR6 scenarios. Although the yield of IR6 was slightly higher ($5.95 \pm 0.14$ kg kg$^{-1}$ versus $5.91 \pm 0.19$ kg kg$^{-1}$), the significant difference between these yields was not recorded due to the honestly significant difference (HSD), which was 0.24 kg kg$^{-1}$ (Figure 6). As mentioned in Table 1, the difference between IR5 and IR6 is the water flow (2 and 3 l min$^{-1}$, respectively) and the total amount of water used per day (16 and 24 L, respectively). Thus, although no significant differences in yield were observed between these scenarios, it would still be worthwhile to choose IR5, since the water consumption is 33% lower compared to IR6. Studies have shown that the same yield but different amounts of water used for irrigation indicate that transpiration processes took place at different rates during growth.

Upon analyzing the research data, a trend was observed between different irrigation frequencies and durations. In hydroponics, the frequency of irrigation has been studied quite widely, but there is no consensus in the scientific area due to the different needs of plants [40–42]. The frequency of irrigation is very important in controlling the concentration of oxygen in the root zone, so that various processes caused by hypoxia do not start [43]. For instance, if wheat sprouts are relatively small and are not able to transpire water from the entire root zone, then the roots may become depleted of oxygen without becoming depleted of water [44]. Under these conditions, more frequent irrigations, but with less water, are needed to provide the plants with needed oxygen. In IR1, IR2, and IR3, with an irrigation duration of 120 s and a frequency of 4 times day$^{-1}$, the average yield reached $4.7 \pm 1.85$ kg kg$^{-1}$, and in modes IR4, IR5, and IR6, with an irrigation duration of 60 s, but the frequency 8 times day$^{-1}$, the average yield was 15.4% higher—$5.55 \pm 1.63$ kg kg$^{-1}$.

The effect of different water flow rates on yield was also observed In the study. Several scientific studies have been conducted on the influence of different irrigation flow rates on the yield or quality of production. Using the hydroponic system and the NFT irrigation fluid delivery system, plants such as lettuce [45–47], Swiss chard [48], water spinach [49], and cauliflower [50] were analyzed in terms of flow rate.

In this study, the water flow rate was the same between IR1 and IR4, between IR2 and IR5, and between IR3 and IR6—only the frequency and duration of irrigation differed. This time, when evaluating the flow rate and amount of water, the highest yield was recorded using scenarios with a water flow rate of 3 l min$^{-1}$ (IR3 and IR6)—$5.72 \pm 0.22$ kg kg$^{-1}$. As the flow rate decreased, the yield was also noticeably lower: for IR1 and IR4 $4.40 \pm 0.38$ kg kg$^{-1}$ and for IR2 and IR5 $5.24 \pm 0.61$ kg kg$^{-1}$. Statistically, the HSD for these parameters was 0.23 kg kg$^{-1}$, so no significant difference was recorded between scenarios with a flow rate of 2 and 3 l min$^{-1}$. However, when choosing the irrigation strategy, it is important to mention that during the empirical tests, a tendency was observed: when the slope angle

of the growth tray was 6.5% or more, and the irrigation flow rate approached $3\,l\,min^{-1}$, in the first days of the growth cycle, seeds were washed to the bottom of the growth tray. Therefore, to avoid this, it is still recommended to choose an irrigation scenario with a flow rate of $2\,l\,min^{-1}$. In terms of yield, an advantage of IR5 and IR6 over the others started to appear from the third day of the growth cycle, and compared to IR3, which had a lower irrigation frequency but longer duration, the advantage was observed only from the fifth day of growing (Figure 7). Although IR6 received a greater amount of water than IR5 during each irrigation, the advantage was not recorded on any days of the growth cycle (Figure 7).

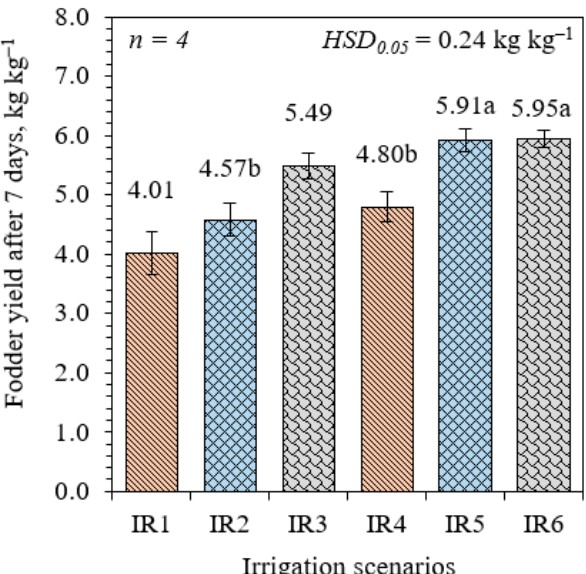

**Figure 6.** Wheat fodder yields after a 7-day growth cycle under different irrigation scenarios. Four repetitions *n* were performed in each scenario. Any two samples with a common letter are not significantly different ($p > 0.05$), as assessed using honestly significant difference.

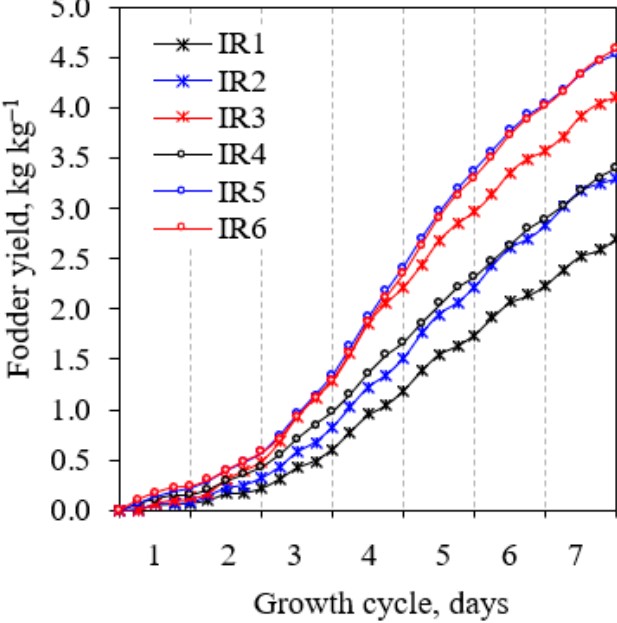

**Figure 7.** Dynamics of wheat fodder yields during a 7-day growth cycle under different irrigation scenarios. Four repetitions *n* were performed in each scenario.

In terms of resource efficiency, the average wheat fodder weight gain per liter of water used over the entire growth cycle also showed which scenario used the water resource most efficiently (Figure 8a). When it comes to hydroponic fodder, the efficient use of water has been studied [6,51], but different flow rates and irrigation frequencies were not investigated. It was found that wheat fodder grown using IR2 produced the highest ($85.64 \pm 4.36$ g kg$^{-1}$) yield with one liter of water used. IR1 produced 16% less—$71.67 \pm 6.55$ g kg$^{-1}$—while significant differences were found only between IR3 and IR6.

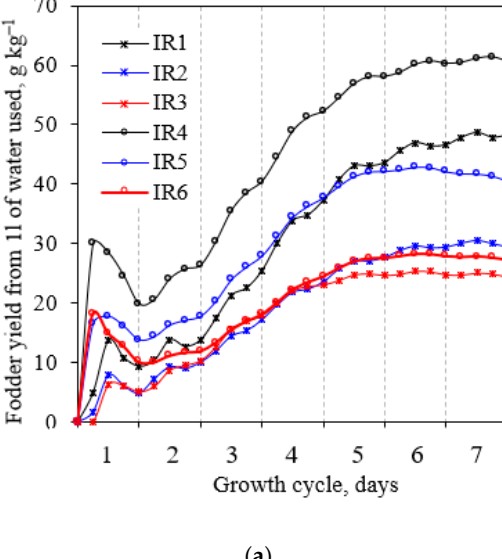

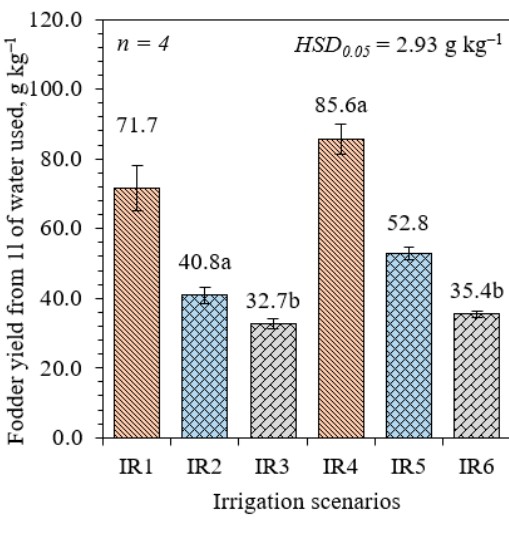

(**a**)  (**b**)

**Figure 8.** (**a**) Dynamics of wheat fodder yields from 1 L of water used during a 7-day growth cycle under different irrigation scenarios. Four repetitions *n* were performed in each scenario. (**b**) Wheat fodder yields from 1 L of water used after a 7-day growth cycle under different irrigation scenarios. Four repetitions *n* were performed in each scenario. Any two samples with a common letter are not significantly different ($p > 0.05$), as assessed using honestly significant difference.

Analyzing the dynamics of wheat fodder yield from 1 L of irrigated water, during a 7-day growth cycle (Figure 8b), it was observed that in different irrigation scenarios, from the first day of cultivation, the highest amount of fodder with 1 L of water was grown in scenario IR4. Interestingly, scenario IR1 was the second in terms of water use efficiency (Figure 8a) and only on the sixth day gained an advantage over IR5, which had a higher frequency of irrigation compared to IR4 and IR1.

It can be concluded that more frequent watering with a smaller amount of water will not only help to grow a higher yield but will also use the water resource more efficiently. However, the aim of this study was not to investigate the efficient use of water in terms of biological processes, i.e., in the cultivation of fodder crops, but to investigate and answer the question of whether the frequency, duration, and amount of water will affect the yield and environmental sustainability. In addition, it would be useful to investigate the influence of different irrigation scenarios by changing the growth tray dimensions (i.e., length and width) and seed layer thickness. It is also important to study the microbiological contamination of this feed, to study different disinfection methods. Although there are studies that studied different disinfectants for seeds [52–54], what influence it would have on the yield of hydroponic fodder and the efficiency of the growth process has not been studied.

### 3.2. Environmental Assessment

The environmental sustainability of growing plants hydroponically has been widely studied [55–58]. Environmental modeling in this study helped to confirm the hypothesis that more frequent irrigation with a lower irrigation flow rate and duration will have the lowest $CO_2$ footprint per FU. It was found that the lowest $CO_2$ footprint is in wheat fodder

that was grown using IR5 and IR6. While the HSD was 6.63 kg $CO_{2eq}$ $t^{-1}$, no significant difference was observed between the two scenarios (Figure 9). Analyzing the irrigation efficiency results, it is already known that IR4 and IR1 had the greatest impact on growing fodder with 1 L of water. However, after analysis, it was found that water consumption accounted for only about 2% of the total global warming potential (GWP). Therefore, growers should pay attention not to water consumption, but to a higher crop yield.

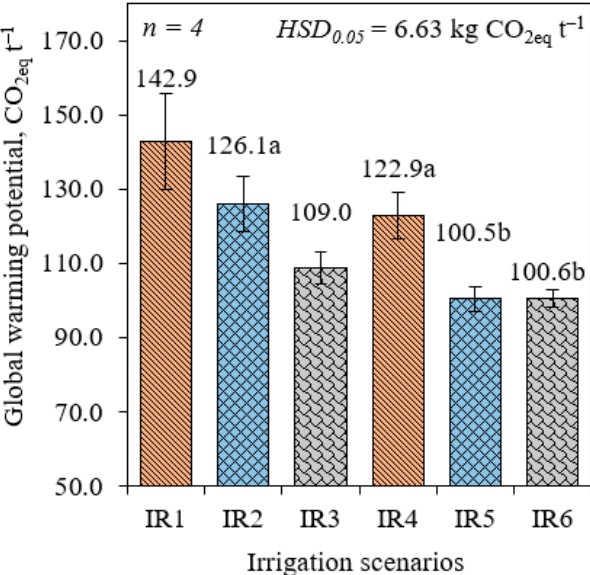

**Figure 9.** The impact of different irrigation scenarios on the environment when growing wheat fodder hydroponically using a 7-day growth cycle. Four repetitions *n* were performed in each scenario. Any two samples with a common letter are not significantly different (*p* > 0.05), as assessed using honestly significant difference.

Comparing the influence of different irrigation frequencies on the $CO_2$ footprint, it was found that when irrigating 4 times $day^{-1}$ (IR1, IR2, and IR3), but with a higher flow rate and duration, the average GWP reaches 126.01 ± 9.74 kg $CO_{2eq}$ $t^{-1}$, and when using more frequent irrigation (8 times $day^{-1}$), only with a lower flow rate and duration, the GWP is 107.98 ± 7.16 kg $CO_{2eq}$ $t^{-1}$. To confirm this, a statistical evaluation was performed, and it was determined that there is a significant difference between these values when the HSD is 11.39 kg $CO_{2eq}$ $t^{-1}$.

Comparing all three modes of irrigation flow rate (1, 2, and 3 l $min^{-1}$), the fodder grown using a 1 l $min^{-1}$ flow rate will have the greatest impact on the environment—the average $CO_2$ footprint is 132.88 ± 10.23 kg $CO_{2eq}$ $t^{-1}$. Irrigation scenarios with flow rates of 2 and 3 l $min^{-1}$ will have the lowest environmental impact (113.30 ± 11.82 and 104.81 ± 4.13 kg $CO_{2eq}$ $t^{-1}$, respectively). However, there is no significant difference between these two scenarios (HSD—11.61 kg $CO_{2eq}$ $t^{-1}$). It can be concluded that in all cases when irrigation is more frequent with a flow rate of 2 or 3 l $min^{-1}$, the wheat fodder is most sustainable in terms of GWP.

As the environmental assessment continued, 11 environmental evaluation criteria were used to accurately assess the impact of different irrigation scenarios on the environment (Table 4). As shown in Table 4, after normalizing the assessment of different types of environmental impact, it was determined which categories of influence are the most important in descending order: marine aquatic ecotoxicity, abiotic depletion (fossil fuels), global warming potential, freshwater aquatic ecotoxicity, human toxicity, terrestrial ecotoxicity, eutrophication, acidification, photochemical oxidation, abiotic depletion (non-fossil fuels), and ozone layer depletion. As in the analysis of the global warming potential, as well as in all the remaining environmental impact categories, the best result is recorded in IR5 and IR6 (Table 4). There is also a trend that more frequent irrigation helps achieve sustainable

fodder production goals. Irrigation scenarios (IR4, IR5, and IR6) whose frequency was 8 times day$^{-1}$ had a 15% lower impact on the environment in all impact categories than the average impact of IR1, IR2, and IR3 combined.

**Table 4.** Environmental performance of different irrigation scenarios based on their midpoint impacts.

| Impact Category | Unit | IR1 | IR2 | IR3 | IR4 | IR5 | IR6 |
|---|---|---|---|---|---|---|---|
| MAE | kg 1,4-DB$_{eq}$ | 44,032.5 | 40,219.7 | 35,902.8 | 37,872.1 | 32,033.2 | 33,134.4 |
| ADf | MJ | 1372.9 | 1213.8 | 1050.4 | 1180.8 | 966.7 | 969.4 |
| GWP | kg CO$_{2eq}$ | 142.8 | 126.1 | 109.0 | 122.8 | 100.4 | 100.6 |
| FWAe | kg 1,4-DB$_{eq}$ | 33.6 | 30.1 | 26.4 | 28.8 | 24.0 | 24.4 |
| HT | kg 1,4-DB$_{eq}$ | 26.8 | 24.2 | 21.3 | 23.0 | 19.2 | 19.6 |
| TE | kg 1,4-DB$_{eq}$ | 1.5 | 1.3 | 1.1 | 1.3 | 1.0 | 1.0 |
| ET | kg PO$_4{}^-$$_{eq}$ | 1.0 | 0.9 | 0.7 | 0.9 | 0.7 | 0.7 |
| ACD | kg SO$_{2eq}$ | 0.8 | 0.7 | 0.6 | 0.7 | 0.6 | 0.6 |
| PO | kg C$_2$H$_{4eq}$ | $6.3 \times 10^{-3}$ | $5.8 \times 10^{-3}$ | $5.2 \times 10^{-3}$ | $5.4 \times 10^{-3}$ | $4.6 \times 10^{-3}$ | $4.8 \times 10^{-3}$ |
| ADn | kg Sb$_{eq}$ | $9.6 \times 10^{-5}$ | $8.9 \times 10^{-5}$ | $8.0 \times 10^{-5}$ | $8.2 \times 10^{-5}$ | $7.0 \times 10^{-5}$ | $7.4 \times 10^{-5}$ |
| ODP | kg CFC-11$_{eq}$ | $4.8 \times 10^{-6}$ | $4.3 \times 10^{-6}$ | $3.7 \times 10^{-6}$ | $4.1 \times 10^{-6}$ | $3.4 \times 10^{-6}$ | $3.4 \times 10^{-6}$ |

Marine aquatic ecotoxicity (MAE) is a criterion for assessing the impact of toxic substances on the marine ecosystem arising from electricity consumption and the emission of non-ferrous metals into the air [59]. Sensitivity analysis showed that in all irrigation scenarios, electricity use accounted for the largest share of MAE, which averaged 86.9 ± 3.3%, and the water consumed was 8.1 ± 3.5%. In addition, MAE accounted for the largest share of total environmental impacts in 7 out of 11 impact categories. To reduce the impact of hydroponic fodder production on the environment, it would be worthwhile to evaluate all sources of electricity, especially renewable ones. The second resource, according to the calculated influence on the environment, was seeds needed for germination. It accounted for 60% of the total GWP, 43% of ADf, and 48% of FWAe. Electricity, water consumption, and seeds needed for germination are the main environmental impact criteria. Although these resources cannot be abandoned due to objective reasons, attention should still be paid to increasing the yield with the lowest possible costs.

## 4. Conclusions

Conducted studies have shown that different irrigation scenarios have a significant impact on the results of hydroponic wheat fodder production. When the production goal is to achieve the highest yield from the seed used, the most suitable scenarios are IR5 (5.95 ± 0.14 kg kg$^{-1}$) and IR6 (5.91 ± 0.19 kg kg$^{-1}$), where the frequency of irrigation is 8 times day$^{-1}$ and the amount of water supplied is 2 and 3 l min$^{-1}$, respectively. Analyzing research data, a trend was observed between different irrigation frequencies and durations. In IR1, IR2, and IR3, with a duration of 120 s and a frequency of 4 times day$^{-1}$, the average yield was 15.4% higher than in IR4, IR5, and IR6 combined, where duration was 60 s, with a frequency of 8 times day$^{-1}$. When evaluating the flow rate and amount of water, the highest yield was recorded using scenarios IR3 and IR6 with a water flow rate of 3 l min$^{-1}$. As the flow rate decreased, the yield was also noticeably lower: IR1 and IR4 by 23% and IR2 and IR5 by 8.4%. In terms of resource efficiency, the average wheat fodder weight gain per liter of water used over the entire growth cycle also showed that wheat fodder grown using IR2 produced the highest fodder yield per liter of water used. IR1 produced 16% less, while significant differences were found only between IR3 and IR6.

Analyzing LCA results, the lowest environmental impact per FU, judging by the amount of fodder obtained from the used seed, was observed in IR5 and IR6. Comparing the influence of different irrigation frequencies on the CO$_2$ footprint, it was found that when irrigating 4 times day$^{-1}$ (IR1, IR2, and IR3), but with a higher flow rate and duration, the average GWP reached 126.01 ± 9.74 kg CO$_{2eq}$ t$^{-1}$, and when using more frequent irrigation, only with a lower flow rate and duration, the GWP was 14.3% lower. In terms of flow rate, in all cases, when irrigation was more frequent with a flow rate of 2 or 3 l min$^{-1}$, the wheat fodder was the most sustainable in terms of GWP. Sensitivity analysis showed

that in all scenarios, electricity use accounted for the largest share of MAE, which averaged $86.9 \pm 3.3\%$.

The research results obtained will help to form recommendations for hydroponic fodder growers on how to use water more economically, in order not only to achieve better economic indicators of the farm but also to contribute to more sustainable use of expensive resources. Future research should focus on the influence of the length of the growth tray on the efficiency of HWF production. Since the quality of irrigation in hydroponics is particularly dependent on the length of the growth tray, a deeper analysis of this aspect is required.

**Author Contributions:** Conceptualization, A.G., D.S. (Dainius Steponavičius), D.S. (Dainius Savickas), Ž.N. and J.B.; methodology, A.G., D.S. (Dainius Steponavičius), D.S. (Dainius Savickas), Ž.N. and J.B.; validation, D.S. (Dainius Steponavičius) and D.S. (Dainius Savickas); investigation, A.G.; resources, A.G.; writing—original draft preparation, A.G. and D.S. (Dainius Savickas); writing—review and editing, A.G., D.S. (Dainius Steponavičius) and D.S. (Dainius Savickas); visualization, A.G.; supervision, D.S. (Dainius Steponavičius). All authors have read and agreed to the published version of the manuscript.

**Funding:** This research received no external funding.

**Data Availability Statement:** Data will be made available on request.

**Conflicts of Interest:** The authors declare no conflict of interest.

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
