# Peer review of "The Influence of Different Irrigation Scenarios on the Yield and Sustainability of Wheat Fodder under Hydroponic Conditions"

_agronomy, doi:10.3390/agronomy13030860_

Round 1
Reviewer 1 Report
Review on „The influence of different irrigation scenarios on the yield and 2 sustainability of wheat fodder under hydroponic conditions”
Abstract
Line 14 etc.: What do IR1, IR2, IR3, IR4, etc. mean? The abstract must be clear and understandable without reading the whole manuscript. The authors should define clearly the used treatments.
Keywords: Please arrange the keywords in alphabetical order.
Introduction:
Why did the authors use wheat as a test plant?
The goal and the hypothesis of this study are not clear. The authors stated that agriculture (crop production) uses lots of water and a new alternative way for crop production is required. This alternative solution would be hydroponic growing conditions (abstract). To prove this statement, a control which is conventional irrigation, would be appropriate. The authors should compare the results of this experiment with the results from conventional irrigation to prove their hypothesis. However, the authors stated other goals in the Introduction section: „1) a higher frequency of irrigation with a lower flow rate and duration will have the most effect on the yield of hydroponic fodder, and (2) fodder grown under these conditions will have the lowest CO2 footprint per functional unit.”
Materials and Methods:
Line 128. Use italics for Latin names.
Line 130: What does the author mean by „tissue paper”? Filter paper? Tissue papers are paper towels, napkins, and toilet paper. I hope that the authors used filter paper for the germination, not simple kitchen towels.
Line 132-134. Delete „There are various methods of grain disinfection: treatment with ozone, electrochemically treated water (anolyte) [28], or hydrogen peroxide [29]”.
Line 135-13. Did the authors first sterilize the seeds with 3% H2O2 and after that the seeds were soaked in tap water? Why? They should use distilled water after the sterilization. If this is an experiment – it is because the authors sterilized the seeds, while the daily cultivation does not need any sterilization process – the authors should be paid more attention to the preparation of the seed.
Line 196. Do not start a sentence with an abbreviation. Please check the whole article.
Could the authors please attach photos of the system? I think that would be more visual together with the draft version.
Results and Discussion:
Please separate the Results and Discussion into two different sections. This is a research article format requirement of Agronomy-MDPI.
Please add the number of repetitions to every table and figure. e.g. n=x, MEAN±S.D. or S.E.
What do lowercase letters mean in tables and figures? The authors should clarify and identify these letters under the tables/figures.
Author Response
Article
The influence of different irrigation scenarios on the yield and sustainability of wheat fodder under hydroponic conditions
Response to Reviewer 1 Comments
We are very grateful to the Reviewer for his good-natured remarks, comments, and suggestions. Please see the point-to-point responses.
Point 1: Line 14 etc.: What do IR1, IR2, IR3, IR4, etc. mean? The abstract must be clear and understandable without reading the whole manuscript. The authors should define clearly the used treatments.
Response 1: We took the reviewer’s suggestions into account and improved our abstract. For clarity, we have added the values of IR1, IR2, IR3, IR4 etc. with explanations (lines 11–16).
Point 2: Keywords: Please arrange the keywords in alphabetical order.
Response 2: We took the reviewer’s suggestions into account and arranged the keywords in alphabetical order (lines 27–28).
Point 3: Introduction: Why did the authors use wheat as a test plant?
Response 3: We took the reviewer’s suggestions into account and improved our materials and methods section. We did this by explaining why wheat was chosen for hydroponic cultivation (lines 136–138).
Point 4: The goal and the hypothesis of this study are not clear. The authors stated that agriculture (crop production) uses lots of water and a new alternative way for crop production is required. This alternative solution would be hydroponic growing conditions (abstract). To prove this statement, a control which is conventional irrigation, would be appropriate. The authors should compare the results of this experiment with the results from conventional irrigation to prove their hypothesis. However, the authors stated other goals in the Introduction section: „1) a higher frequency of irrigation with a lower flow rate and duration will have the most effect on the yield of hydroponic fodder, and (2) fodder grown under these conditions will have the lowest CO2 footprint per functional unit.”.
Response 4: Thank you for your insight, however we did not claim that the goal of our study is to compare conventional agriculture with hydroponic technology in terms of fodder production. As mentioned in the article, the advantage of hydroponics in terms of water use efficiency over traditional agriculture has already been proven. The purpose of this article was to prove that hydroponic fodder production can be made even more efficient. Also, to confirm the proposed hypotheses using empirical research methods.
Point 5: Line 128. Use italics for Latin names.
Response 5: We took the reviewer’s suggestions into account and used italcs for Latin names (line 139).
Point 6: Line 130: What does the author mean by „tissue paper”? Filter paper? Tissue papers are paper towels, napkins, and toilet paper. I hope that the authors used filter paper for the germination, not simple kitchen towels.
Response 6: We took the reviewer’s suggestions into account and replaced “tissue” with “filter” (line 141). Filter paper was indeed used in our research.
Point 7: Line 132-134. Delete „There are various methods of grain disinfection: treatment with ozone, electrochemically treated water (anolyte) [28], or hydrogen peroxide [29]”.
Response 7: We took the reviewer’s suggestions into account and deleted „There are various methods of grain disinfection: treatment with ozone, electrochemically treated water (anolyte) [28], or hydrogen peroxide [29]”.
Point 8: Line 135-13. Did the authors first sterilize the seeds with 3% H2O2 and after that the seeds were soaked in tap water? Why? They should use distilled water after the sterilization. If this is an experiment – it is because the authors sterilized the seeds, while the daily cultivation does not need any sterilization process – the authors should be paid more attention to the preparation of the seed.
Response 8: Thank you for your observations. However, our study was designed to replicate the production conditions found in our region. We understand your concern about using tap water after disinfection. In hydroponic fodder production factories located in Lithuania, tap water is used for soaking and watering seeds, since, according to the current sanitary regulations (HN 24:2003) of our region, tap water cannot contain any mycotoxins or various microorganisms, therefore it is safe for human and animal consumption.
Indeed, we would not have thought that such a question might arise for the reader, therefore, according to the question you raised, we made certain corrections in the text, thereby bringing clarity as to why tap water was chosen as an irrigation solution (lines 145–148).
Point 9: Line 196. Do not start a sentence with an abbreviation. Please check the whole article.
Response 9: The reviewer's suggestions have been taken into account and we have checked the entire article to avoid starting a sentence with an abbreviation.
Point 10: Could the authors please attach photos of the system? I think that would be more visual together with the draft version.
Response 10: We took the reviewer’s suggestions into account and attached two photos of the system.
Point 11: Please separate the Results and Discussion into two different sections. This is a research article format requirement of Agronomy-MDPI.
Response 11: Thank you for your comment, but according to the guidelines, results and discussion can be written in one paragraph. We quote: "Authors should discuss the results and how they can be interpreted in perspective of previous studies and of the working hypotheses. The findings and their implications should be discussed in the broadest possible context and limitations of the work highlighted. Future research directions may also be mentioned. This section may be combined with Results.” Source - https://www.mdpi.com/journal/agronomy/instructions.
Considering your comment, in addition, we expanded the discussion to include limitations of our study and potential directions (literature 53–55) for future research (lines 340–346).
Point 12: Please add the number of repetitions to every table and figure. e.g. n=x, MEAN±S.D. or S.E.
Response 12: We have considered the reviewer's suggestions and added the number of repetitions in the figures where necessary. Standard errors are given in the text to avoid overloading the graphs, making it easier for the reader.
Point 13: What do lowercase letters mean in tables and figures? The authors should clarify and identify these letters under the tables/figures.
Response 13: We took the reviewer’s suggestions into account and clarified “lowercase letters” under the figures where statistical analysis was performed.

Reviewer 2 Report
The article "The influence of different irrigation scenarios on the yield and sustainability of wheat fodder under hydroponic conditions" is well written. The English language is at a good level and allows a full understanding of the work. The authors precisely defined the study's purpose and included the research hypotheses. The methodology was presented accurately, and diagrams and figures facilitated understanding of the entire experiment. The results were presented correctly in many graphs. Moreover, in the last chapter, the authors detailed their research's main conclusions and indicated the directions for future experiments. I believe that this manuscript is suitable for publication in the journal Agronomy after making a few minor corrections:
In the abstract, the authors write that the best yield results were achieved with the IR5 and IR6 scenarios; however, it is unclear what is behind these abbreviations. Therefore, please describe the scenario briefly to make the abstract understandable to the reader without reading the entire text. The same applies to the other scenarios described in the abstract.
Even though the NFT system is known to those in the industry, I would suggest expanding its description so that people who are not involved in the subject daily understand the idea of hydroponics more.
The authors could expand the discussion based on more articles published on this topic. This would be of value to the article.
Editorial comments:
In line 106, there should be irrigation not rrigation.
In line 234, there are two dots at the end. Remove one.
In line 287, -1 should be in superscript.
Author Response
Article
The influence of different irrigation scenarios on the yield and sustainability of wheat fodder under hydroponic conditions
Response to Reviewer 2 Comments
We are very grateful to the Reviewer for his good-natured remarks, comments, and suggestions. Please see the point-to-point responses.
Point 1: In the abstract, the authors write that the best yield results were achieved with the IR5 and IR6 scenarios; however, it is unclear what is behind these abbreviations. Therefore, please describe the scenario briefly to make the abstract understandable to the reader without reading the entire text. The same applies to the other scenarios described in the abstract.
Response 1: We took the reviewer’s suggestions into account and improved our abstract. For clarity, we have added the values of IR1, IR2, IR3, IR4 etc. with explanations (lines 11–16).
Point 2: Even though the NFT system is known to those in the industry, I would suggest expanding its description so that people who are not involved in the subject daily understand the idea of hydroponics more.
Response 2: We took the reviewer’s suggestions into account and a brief explanation of the NFT system has been added to make things easier for the reader (lines 54–58).
Point 3: The authors could expand the discussion based on more articles published on this topic. This would be of value to the article.
Response 3: We have considered the reviewer's suggestions and expanded the discussion to include limitations of our study and potential directions (literature 53–55) for future research (lines 340–346).
Point 4: In line 106, there should be irrigation not rrigation. In line 234, there are two dots at the end. Remove one. In line 287, -1 should be in superscript.
Response 4: We took the reviewer’s suggestions into account and fixed mentioned issues regarding editorial comments.

Reviewer 3 Report
The authors report in this paper the results of an alternative to solve the problem of waste of water used in agriculture. Hydroponically grown animal feed, which under hydroponic conditions can achieve optimal results and, above all, use water more efficiently. In the research conducted by the authors, different irrigation scenarios were studied. The research work presented by the authors remains original. In terms of plagiarism assessment, only about 15% of this document consists of text more or less similar to the content of the sources considered most relevant of iThenticate. The research topic is very significant for the field. The topic is interesting, and the methodology is appropriate for this category of research. The manuscript is written in a very clear and interesting way. It is well organized and structured. The results are clearly presented. References are cited as per the authors' instructions. Overall, I recommend the publication of this paper in Agronomy in its present form.

Author Response
ArticleThe influence of different irrigation scenarios on the yield and sustainability of wheat fodder under hydroponic conditions
Response to Reviewer 3 Comments
We are very grateful to the Reviewer for his comments and kind words
Andrius Grigas
Round 2
Reviewer 1 Report
The authors completed most of my suggestions. However, one serious point maybe was not understandable. Please, indicate the number of repetitions of each measured parameter in the figures or tables’ title. I was looking for exact numbers not for „n=number of repetitions”, e.g. n=3, n=5, etc.
